# High-Sensitivity pH Sensor Based on Coplanar Gate AlGaN/GaN Metal-Oxide-Semiconductor High Electron Mobility Transistor

**Seong-Kun Cho and Won-Ju Cho \***

Department of Electronic Materials Engineering, Kwangwoon University, 20 Gwangun-ro, Nowon-gu, Seoul 01897, Korea; whtjdrms98@gmail.com

\* Correspondence: chowj@kw.ac.kr; Tel.: +82-2-940-5163

**Abstract:** The sensitivity of conventional ion-sensitive field-effect transistors is limited to the Nernst limit (59.14 mV/pH). In this study, we developed a pH sensor platform based on a coplanar gate AlGaN/GaN metal-oxide-semiconductor (MOS) high electron mobility transistor (HEMT) using the resistive coupling effect to overcome the Nernst limit. For resistive coupling, a coplanar gate comprising a control gate (CG) and a sensing gate (SG) was designed. We investigated the amplification of the pH sensitivity with the change in the magnitude of a resistance connected in series to each CG and SG via Silvaco TCAD simulations. In addition, a disposable extended gate was applied as a cost-effective sensor platform that helped prevent damages due to direct exposure of the AlGaN/GaN MOS HEMT to chemical solutions. The pH sensor based on the coplanar gate AlGaN/GaN MOS HEMT exhibited a pH sensitivity considerably higher than the Nernst limit, dependent on the ratio of the series resistance connected to the CG and SG, as well as excellent reliability and stability with non-ideal behavior. The pH sensor developed in this study is expected to be readily integrated with wide transmission bandwidth, high temperature, and high-power electronics as a highly sensitive biosensor platform.

**Keywords:** AlGaN/GaN MOS HEMT; coplanar gate; resistive coupling; pH sensor

## 1. Introduction

Developments in big data, artificial intelligence, deep learning, and internet of things in the fourth industrial revolution have led to increased human–machine interactions and, consequently, the requirement of the miniaturization and functionalization of electronic devices. In particular, biosensors are being actively studied as a core technology for human–machine interactions. Field-effect transistor (FET)-based biosensors offer various advantageous properties, including a short response time and accurate detection, and they can be mass produced and miniaturized [1,2]. However, conventional ion-sensitive FETs (ISFETs) have a sensitivity limit of 59.14 mV/pH, called the Nernst limit, which hinders their commercialization [3,4]. Double gate structure ISFETs based on silicon-on-insulator are drawing attention because they can overcome the Nernst limit via asymmetric capacitive coupling of the upper and lower gate oxides [5–8]. In addition, in our previous work, we studied a silicon-based device that amplifies the sensitivity through capacitive coupling between the coplanar gate and the floating gate, and here we investigated the capacitive coupling effect according to the coplanar gate area and the corresponding sensitivity amplification [9]. However, owing to the limitations of the physical characteristics of silicon, these transistors cannot easily be integrated with wide transmission bandwidths and at high temperatures. Meanwhile, compound semiconductor-based AlGaN/GaN high electron mobility transistors (HEMTs) exhibit low on-resistance and rapid switching speeds owing to the high-mobility two-

dimensional electron gas induced by the discontinuity of the conduction band in the AlGaN/GaN heterostructure and its piezoelectric polarization [10]. In addition, AlGaN/GaN HEMT-based sensors have shown potential for application for the detection of DNA, antigens, glucose, cellular responses, and gas; further, these sensors can be monolithically integrated with signal processing and amplification circuits or radio frequency (RF) signal transmission circuits, which are convenient for reading data and receiving remote signals [11–18]. Nevertheless, the leakage current of AlGaN/GaN HEMTs due to the trap-assisted tunneling and surface state is crucial. In order to attain low power consumption and high conversion efficiency, high breakdown voltage, leakage suppression, and high on/off current ratio are crucial. Meanwhile, the metal-oxide-semiconductor (MOS) structure can achieve a high breakdown voltage and on/off current ratio because the gate insulator effectively prevents the gate leakage and suppresses the surface leakage [19,20].

In this study, we developed a pH sensor based on an AlGaN/GaN MOS HEMT using the resistive coupling of a coplanar gate structure comprising a control gate (CG) and a sensing gate (SG). Previous AlGaN/GaN MOS HEMT-based ISFETs can easily be damaged, because the HEMT device in these is directly exposed to a chemical solution. To prevent this damage, we designed an AlGaN/GaN MOS HEMT transducer unit that converts biochemical signals into electrical signals and an extended gate (EG) sensing unit that is directly exposed to the pH solution. This separate structure of the transducer and sensing units fundamentally prevents damage to the expensive HEMT and provides a cost-effective sensor platform by easily replacing the damaged low-cost sensing unit. The variation in the sensitivity of the pH sensor was evaluated according to the ratio of the series resistance connected to the CG ($R_{CG}$) and SG ($R_{SG}$) of the AlGaN/GaN MOS HEMT transducer, and an amplified sensitivity that exceeded the Nernst limit was obtained. In addition, non-ideal behaviors, such as hysteresis and drift effects, were investigated to verify the stability and reliability of the sensor.

## 2. Materials and Methods

The coplanar gate AlGaN/GaN MOS HEMT transducer unit was fabricated as follows. AlGaN/GaN heterostructures were grown in metal organic chemical vapor deposition reactors on (0001) sapphire substrates. A 25 nm low-temperature GaN nucleation layer was grown on the sapphire substrate, followed by the successive growth of a 1 μm high-resistance GaN layer and 1.8 μm unintentionally doped GaN epitaxial layer. Subsequently, a 12 nm $Al_{0.25}GaN$ layer was grown. Mesa separation was realized by etching a 500 nm AlGaN/GaN layer, where the active region was defined by $BCl_3$- and $Cl_2$-based inductively coupled plasma etching using a photoresist (PR) mask. A 500 nm $SiO_2$ field oxide layer was deposited via RF magnetron sputtering for a flat surface to avoid breaking of the gate metals at the steep steps in the mesa region. After removing the PR mask and $SiO_2$ layer from the active region, the AlGaN/GaN heterostructure was cleaned using an ammonium sulfide solution with excess sulfur [$(NH_4)_2S_x$, 40% sulfur] to remove the native oxide and prevent the formation of oxide. Subsequently, a 5 nm $SiO_2$ layer and 10 nm $Ta_2O_5$ layer were deposited as gate insulators via RF magnetron sputtering. The source and drain electrodes of Ti/Al/Ni/Au (=20/100/25/50 nm) were formed via e-beam evaporator and lift-off, followed by rapid thermal treatment at 800 °C for 30 s in $N_2$ ambient. To form series resistors between the HEMT gate (on channel) and the coplanar gates (CG and SG), a 50 nm indium-tin-oxide (ITO) film with a sheet resistance of $1.7 \times 10^3$ Ω/sq was deposited. We designed resistors with a width of 2 μm and lengths of 10, 20, and 30 μm and obtained resistances of 17, 34, and 51 kΩ, respectively. For the HEMT gate and coplanar gate electrodes, Ni/Au (=25/100 nm) was deposited via e-beam evaporation and lift-off. Meanwhile, to fabricate the EG sensing unit, a 150 nm ITO conductive layer and a 50 nm $SnO_2$ sensing membrane were subsequently deposited on glass substrates via RF magnetron sputtering. Subsequently, a

polydimethylsiloxane reservoir was placed on top. Figure 1a,b shows schematics of the coplanar gate AlGaN/GaN MOS HEMT and EG, respectively.

We finally constructed the pH sensor by connecting the EG to the SG of the AlGaN/GaN MOS HEMT with an electrical cable (Model 5342, Pomana electronics, Everett, WA, USA) to transfer the potential of the pH solution to the SG. We performed all electrical measurements of device characteristics using an Agilent 4156B precision semiconductor parameter analyzer (Agilent Technologies, Santa Clara, CA, USA). Also, these measurements were performed in a shielded dark box to prevent external elements such as noise and light. In particular, a commercial Ag/AgCl reference electrode (Horiba 2086A-06T, Kyoto, Japan) composed of ceramic-plug junction and internal solution saturated with KCl and AgCl was used for pH sensing.

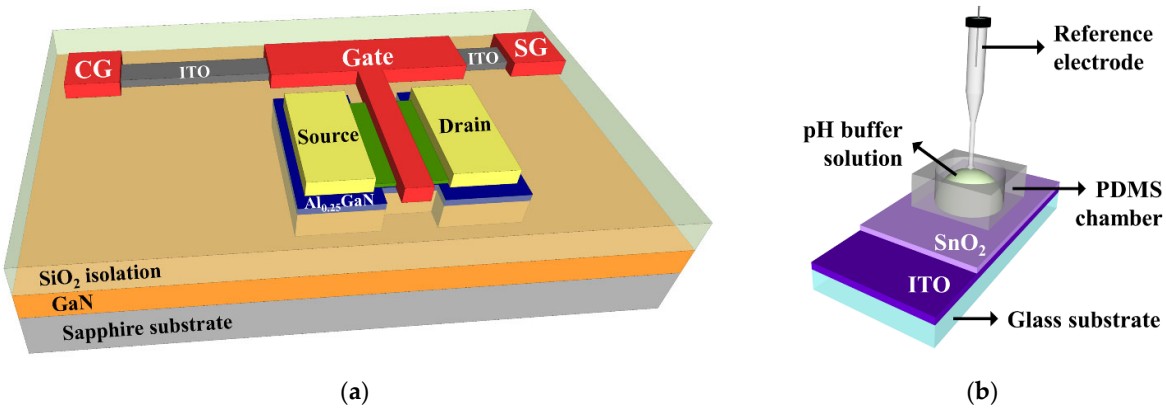

(**a**)                                             (**b**)

**Figure 1.** Schematic of (**a**) coplanar gate AlGaN/GaN metal-oxide-semiconductor high electron mobility transistor (MOS HEMT) transducer unit, and (**b**) extended gate (EG) sensing unit.

## 3. Results

### 3.1. Silvaco TCAD Simulations

The voltages applied to the CG and SG are denoted as $V_{CG}$ and $V_{SG}$, respectively. The total resistance ($R_T$) of a coplanar gate can be expressed as $R_T = R_{CG} + R_{SG}$, where $R_{CG}$ and $R_{SG}$ are the resistances of the CG and SG, respectively. Then, the gate voltage ($V_{FG}$) of the MOS HEMT is expressed as in Equation (1), and the voltage between CG and SG can be calculated using Equation (2). Eventually, the potential change in SG ($\Delta V_{SG}$) is amplified by a factor of $R_{CG}/R_{SG}$, resulting in a voltage change in the CG ($\Delta V_{CG}$), which implies that a significantly small change in potential in the SG is amplified by the resistive coupling effect and can be detected in the CG.

$$V_{FG} = \frac{R_{SG}}{R_T} V_{CG} + \frac{R_{CG}}{R_T} V_{SG}, \tag{1}$$

$$V_{CG} = \frac{R_T}{R_{SG}} V_{FG} - \frac{R_{CG}}{R_{SG}} V_{SG}, \tag{2}$$

$$\therefore \Delta V_{CG} \propto \frac{R_{CG}}{R_{SG}} \Delta V_{SG}, \tag{3}$$

Figure 2b shows Silvaco TCAD simulations for the transfer characteristic curves of the AlGaN/GaN MOS HEMT with $R_{CG}:R_{SG}$ = 2:1. The Silvaco TCAD simulation proposed in this study was to evaluate the signal amplification capability by resistive coupling, and was performed as follows: (1) Atals simulation for MOS HEMT device with conmob (specify concentration dependent mobility model), fldmob (specify lateral electric field-dependent model), SRH (specifies Shockley-Read-Hall recombination). (2) Mixedmode module in Atlas for circuit simulation for resistive coupling between CG and SG. (3) Tunneling conduction such as band-to-band tunneling, Fowler-Nordheim tunneling,

direct quantum tunneling, trap assist tunneling, phonon assist electron tunneling, Schottky tunneling, etc. were ignored. As a result, using Equation (3), when $\Delta V_{SG}$ is 2 V, $\Delta V_{CG}$ is calculated to be 4 V, and the resistance ratio $R_{CG}/R_{SG}$ is increased to 2. Figure 2c shows the dependence of the amplification factor ($\Delta V_{CG}/\Delta V_{SG}$) considering ratios $R_{CG}$:$R_{SG}$ = 1:2, 1:1, 2:1, and 3:1, where $\Delta V_{CG}$ is determined at a drain current of 1 nA. The amplification coefficients at ratios $R_{CG}$:$R_{SG}$ = 1:2, 1:1, 2:1, and 3:1 were found to be 0.5, 1, 2, and 3, respectively.

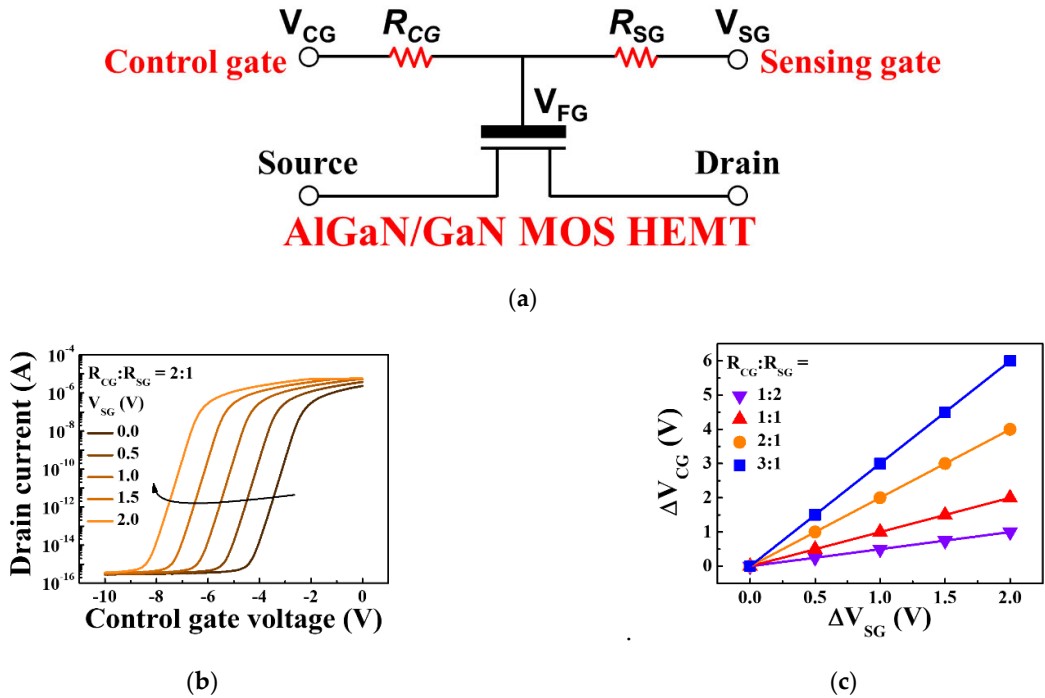

**Figure 2.** (**a**) Simplified equivalent circuit of the pH sensor based on the coplanar gate AlGaN/GaN MOS HEMT proposed herein. Silvaco TCAD simulation results of (**b**) transfer characteristic curves shift as a function of $\Delta V_{SG}$, and (**c**) dependence of the amplification factor ($\Delta V_{CG}/\Delta V_{SG}$) on $R_{CG}$:$R_{SG}$ of the pH sensor.

### 3.2. Electrical Characteristics of Coplanar Gate AlGaN/GaN MOS HEMT

Figure 3a shows the transfer characteristic curves of the fabricated coplanar gate AlGaN/GaN MOS HEMT. Here, the threshold voltage is −0.99 V, electron concentration is $7.47 \times 10^{12}$ cm$^{-2}$, electron mobility is 2835.9 cm$^2$/V s, on/off current ratio is $5.55 \times 10^8$, and subthreshold swing is 79.08 mV/dec. A low gate leakage current was observed because the $SiO_2$/$Ta_2O_5$ stacked gate insulator effectively blocked the gate leakage and suppressed the surface leakage. In addition, the output characteristic curves in Figure 3b indicate a high drain drive current due to the high electron concentration and mobility of the AlGaN/GaN MOS HEMT.

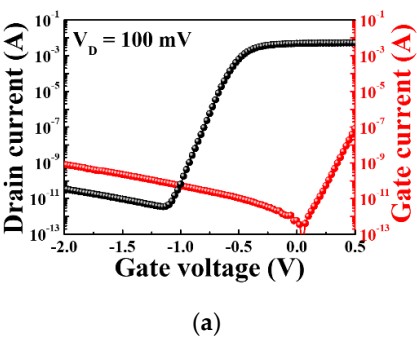
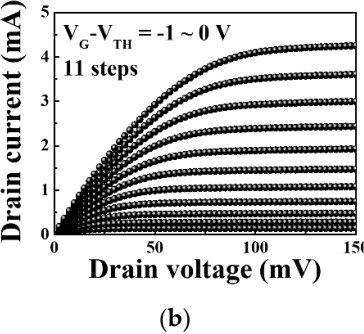

**Figure 3.** (**a**) Transfer characteristics and (**b**) output characteristic curves of the fabricated AlGaN/GaN MOS HEMT.

### 3.3. pH Sensing Characteristics of Coplanar Gate AlGaN/GaN MOS HEMT

There is a Nernst limit for pH sensing properties that cannot exceed 59.14 mV/pH according to site-binding theory. In the site-binding theory, the ability to detect ions depends entirely on the surface potential ($\psi$), which is summarized in the Equation (4): [21,22]

$$\psi = 2.303 \frac{kT}{q} \frac{\beta}{\beta+1} \left( pH_{pzc} - pH \right), \tag{4}$$

where $k$ is Boltzmann constant, $T$ is the absolute temperature, $q$ is the elementary charge, $pH_{pzc}$ is the pH at the point of zero charge, and $\beta$ is a parameter that denotes the chemical sensitivity of the sensing membrane. The $\psi$ depends on the properties of the sensing membrane and the pH level of the electrolyte. Thus, the shift of the threshold voltage ($\Delta V_{TH}$) for the transistor is determined by $\Delta \psi$. Based on the simulation result, when resistive coupling is introduced, $\Delta V_{TH}$ becomes $R_{CG}/R_{SG}$ times $\Delta \psi$, which can amplify the sensitivity.

Figure 4a–c shows the transfer characteristic curves of the pH sensor for the ratios $R_{CG}$:$R_{SG}$ = 1:1, 2:1, and 3:1, respectively. We measured the sensing properties using buffer solutions with the following pH concentrations: 3.07, 4.08, 5.99, 6.95, 8.97, and 9.87. From Figure 4, it can be seen that as the pH value increases, the transfer characteristic curve shifts in the positive voltage direction. Figure 4d shows a plot of $\Delta V_{CG}$ as a function of the pH value. Here, $\Delta V_{CG}$ is defined as the shift of the $V_{CG}$ extracted from the drain current of 1 nA, corresponding to the pH sensitivity as a function of the concentration of the buffer solution. For $R_{CG}$:$R_{SG}$ = 1:1, the pH sensitivity is 56.63 mV/pH, which is lower than the Nernst limit of 59.14 mV/pH. However, for $R_{CG}$:$R_{SG}$ = 2:1 and 3:1, the pH sensitivities are 112.17 and 167.71 mV/pH, respectively, which are amplified by a factor of $R_{CG}/R_{SG}$ and exceed the Nernst limit. In addition, the linear fitting line for extracting the sensitivity in Figure 4d showed more than 99.8% linearity in $R_{CG}$:$R_{SG}$ = 1:1, 2:1, and 3:1. Figure 4d also shows the deviation for 20 measurements at each pH value, indicating reliable sensing performance. This means that there is no Debye screening length limit for the various pH buffer solutions used in this study.

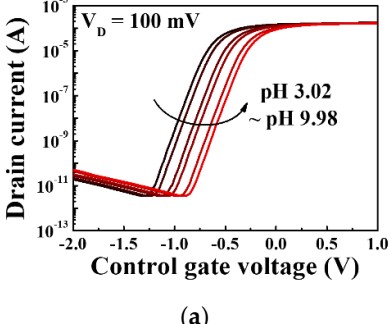
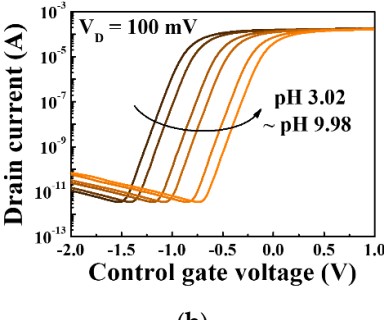

(**a**)                                                         (**b**)

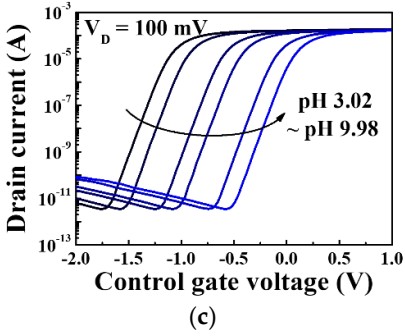
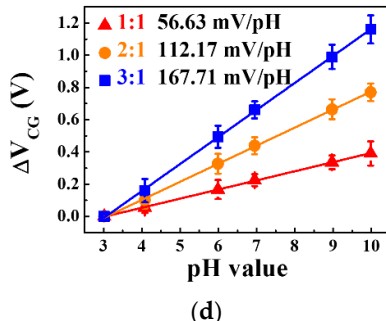

(**c**)       (**d**)

**Figure 4.** Transfer characteristic curves of the pH sensor based on a coplanar gate AlGaN/GaN MOS HEMT for $R_{CG}$:$R_{SG}$ = (**a**) 1:1, (**b**) 2:1, and (**c**) 3:1. (**d**) $\Delta V_{CG}$ as a function of the pH value.

Reliability-related hysteresis effects are due to the presence of slowly reacting OH-sites on the surface of the sensing membrane and the transport of certain species in the bulk of the sensing membrane [23–25]. Meanwhile, stability-related drift effects are due to the hopping and/or trap-limited transport of OH-related species from the electrolyte and defects present in the sensing membrane [26,27]. These non-ideal behaviors limit the accuracy of the sensor. Figure 5a shows the hysteresis characteristics when the pH is changed as follows: 7 → 10 → 7 → 4 → 7. It can be seen that, for $R_{CG}$:$R_{SG}$ = 1:1, 2:1, and 3:1, the hysteresis voltages are 1.8, 3.5, and 7.1 mV, respectively. Figure 5b shows the drift rates upon exposure to a pH 7 buffer solution for 10 h. It can be seen that, for $R_{CG}$:$R_{SG}$ = 1:1, 2:1, and 3:1, the drift rates are 0.4, 0.6, and 0.7 mV/h, respectively.

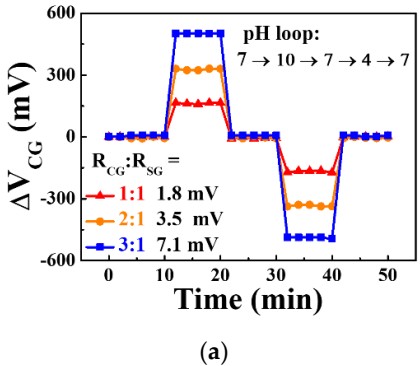
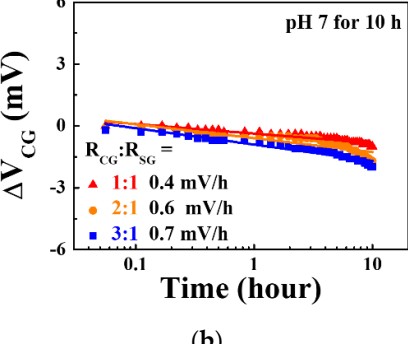

(**a**)       (**b**)

**Figure 5.** (**a**) Hysteresis effect and (**b**) drift effect of the pH sensor based on a coplanar gate AlGaN/GaN MOS HEMT for $R_{CG}$:$R_{SG}$ = 1:1, 2:1, and 3:1.

### 4. Conclusions

We developed a pH sensor based on a coplanar gate AlGaN/GaN MOS HEMT with increased sensitivity using the resistive coupling effect. Through Silvaco TCAD simulations, the pH sensitivity amplification corresponding to the ratio of the series resistance connected to the CG and SG ($R_{CG}/R_{SG}$) is predicted. The pH sensitivity of the fabricated sensors increases proportionally with $R_{CG}/R_{SG}$. In particular, for $R_{CG}$:$R_{SG}$ = 2:1 and 3:1, the sensitivity was 112.17 and 167.71 mV/pH, respectively, which are greater than the Nernst limit. We also verified the stability and reliability of the sensor by evaluating its non-ideal behaviors, such as the hysteresis and drift effects. In addition, by applying the disposable EG sensing unit, damage to the AlGaN/GaN MOS HEMT transducer unit due to chemical solutions was prevented. The results of this study indicate that the pH sensor developed has high performance, stability, and reliability and is also disposable. Thus, since the sensor proposed in this study is based on a HEMT device, it is expected to be suitable for integration with wide transmission bandwidth, high temperature, speed

and frequency power electronics. In addition, the pH sensor proposed in this study detects the potential change of the sensing membrane using EG. Therefore, when the coplanar gate AlGaN/GaN MOS HEMT proposed in this study is introduced into the subsequent study on EG, various biological events such as enzyme-substrate reactions, antigen-antibody binding, and nucleic acid hybridization can be detected with high sensitivity.

**Author Contributions:** S.-K.C.: conceptualization, formal analysis, methodology, investigation, data curation, visualization, software, resources, writing—original draft. W.-J.C.: conceptualization, methodology, investigation, resources, formal analysis, funding acquisition, supervision, validation, writing—review and editing. All authors have read and agreed to the published version of the manuscript.

**Funding:** The present research has been conducted by the Research Grant of Kwangwoon University in 2020, and this work was supported by the National Research Foundation of Korea (NRF) grant funded by the Korea government (MSIT) (No. 2020R1A2C1007586). This research was funded and conducted under the Competency Development Program for Industry Specialists of the Korean Ministry of Trade, Industry and Energy (MOTIE), operated by Korea Institute for Advancement of Technology (KIAT) (No. P0002397, HRD program for Industrial Convergence of Wearable Smart Devices).

**Conflicts of Interest:** The authors declare no conflict of interest.

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
