# Peer review of "High-Sensitivity pH Sensor Based on Coplanar Gate AlGaN/GaN Metal-Oxide-Semiconductor High Electron Mobility Transistor"

_chemosensors, doi:10.3390/chemosensors9030042_

Round 1

Reviewer 1 Report

  1. The author mentioned that electrical cable was used to connect the EG to the SG. What electrical cable was used? Did it increase the noise or maybe drift?
  2. The author mentioned the chemical might damage the electrode surface. But it would be better to compare the results from the SG directly and the EG.
  3. In principle, the method presented here did not overcome the Nernst limit, it utilised the RCG/RSG ratio to amplify the potential. My worry is it amplifies the noise as well. It would be better to see the comparison of the signal to noise ratio among those ratios.
  4. the author presented 4 different RCG/RSG ratios. It seems the higher the ratio, the better the signal. Did the author try a ratio higher than 3:1? Is there any limitation to that? 
  5. What buffer was used for the measurements? Did the author experience Debye screening length limit? 
  6. Is there sensor to sensor variation?
  7. The author mentioned that "the sensor is suitable for integration with wide transmission bandwidth, high temperature, speed, and frequency power electronics.", but no direct evidence or explanation was shown in the article.

Author Response

Response Letter to Reviewer #1
We sincerely thank you for giving us such valuable suggestions for revision. The reviewers′ comments are indeed constructive and helpful for us to improve our manuscript. Thus, we revise our manuscript based on the reviewers′ comments exactly. The followings are our point-to-point response to the reviewers′ concerns and our descriptions on the revision, which are indicated by yellow highlight in the revised manuscript: 

[Comment 1]
The author mentioned that electrical cable was used to connect the EG to the SG. What electrical cable was used? Did it increase the noise or maybe drift?

[Answer 1]
Thank you for your helpful comments. We used a commercial Triax-to-Aligator clip electrical cable (Model 5342, Pomana Electronics, USA). The cable was in direct contact with the transducer's sensing gate through an extended gate and probe in a shielded dark box. Through this connection, we minimized external influences such as noise signals and light.

In response to the reviewer’s comments, we have also revised the manuscript in the following section for the reader's understanding:

(Line 94, Page 2 – Line 101, Page 3 in revised manuscript)
We finally constructed the pH sensor by connecting the EG to the SG of the Al-GaN/GaN MOS HEMT with an electrical cable (Model 5342, Pomana electronics, USA) to transfer the potential of the pH solution to the SG. We performed all electrical measure-ments of device characteristics using an Agilent 4156B precision semiconductor parame-ter analyzer. Also, these measurements were performed in a shielded dark box to prevent external elements such as noise and light. In particular, a commercial Ag/AgCl reference electrode (Horida 2086A-06 T) composed of ceramic-plug junction and internal solution saturated with KCl and AgCl was used for pH sensing.

[Comment 2]
The author mentioned the chemical might damage the electrode surface. But it would be better to compare the results from the SG directly and the EG.

[Answer 2]
There seems to be a misunderstanding by reviewers about our device structure and operating principle. In the sensor presented in this study, CG and SG electrodes are basically not directly exposed to chemicals. In contrast, in the case of a conventional ISFET structure, chemical damage occurs when the sensing layer on the channel is directly exposed. To prevent this, we introduced EG in this study. In other words, the electrical signal of the EG is only transmitted to the SG of the transducer through a low-noise electrical cable. If the EG is damaged, we can solve the problem by simply replacing it with a new one. 

[Comment 3]
In principle, the method presented here did not overcome the Nernst limit, it utilized the RCG/RSG ratio to amplify the potential. My worry is it amplifies the noise as well. It would be better to see the comparison of the signal to noise ratio among those ratios.

[Answer 3]
We admired the sharp and helpful comments. In principle, our proposed method does not overcome the Nernst limit, but amplifies potential by the RCG/RSG ratio, as the reviewer commented. To investigate the effects of noise, a 0.2 V amplitude square wave pulse (assuming noise) was applied to the CG voltage to monitor the drain current fluctuations with the three RCG:RSG ratios shown in Figure R1. In this test, an EG injected with a pH 7 buffer solution was connected to the SG. As a result, the RCG:RSG of 1:1, 2:1, and 3:1 showed similar fluctuations in drain current. In addition, as a result of comparing the maximum (Imax) and minimum (Imin) ratio of the current ratio for quantitative analysis, Imax/Imin = 77.3, 78.4, and 81.3 for the RCG:RSG = 1:1, 2:1, and 3:1, respectively. Therefore, in the AlGaN/GaN MOS HEMT-based pH sensor presented in this paper, we verified that the sensitivity was amplified as the RCG:RSG ratio increased, but the effect of noise was not amplified and almost maintained.

Figure R1. The drain current fluctuation due to 0.2 V amplitude square wave pulse (assuming noise) applied to the control gate voltage.

[Comment 4]
The author presented 4 different RCG/RSG ratios. It seems the higher the ratio, the better the signal. Did the author try a ratio higher than 3:1? Is there any limitation to that?

[Answer 4]
Thanks for your valuable comments. As the reviewer noted, increasing the RCG:RSG ratio increases the threshold voltage shift, which further amplifies the sensitivity. However, in the case of the AlGaN/GaN MOS HEMT used in this study, RCG:RSG patterns of 1:1, 2:1, 3:1 ratio exist in our designed photomask. Unfortunately, we couldn't evaluate the ratio beyond, but in principle, the larger the RCG:RSG ratio, the higher the sensitivity is expected.

[Comment 5]
What buffer was used for the measurements? Did the author experience Debye screening length limit?

[Answer 5]
We measured the sensing characteristics using commercial pH buffer solutions (SAMCHUN CHEMICAL, Republic of Korea). These buffer solutions consist of potassium hydrogen phthalate. In general, the Debye screening length depends on the electrolyte concentration. The lower the electrolyte concentration, the larger the Debye screening length and consequently the interfacial capacitance decreases [R1]. On the other hand, in the VCG change according to the pH value shown in Figure 4(d) of the revised manuscript, the RCG:RSG of 1:1, 2:1, and 3:1 showed higher linearity than 99.8%. In addition, Figure 4(d) shows the deviation for 20 measurements at each pH value, indicating reliable detection performance. Therefore, we consider that there is no limitation on the length of Debye screening for the pH buffer solutions used in this study.

In response to the reviewer’s comments, we have also revised the manuscript in the following section for the reader's understanding:

(Line 163 – 167, Page 5 in revised manuscript)
In addition, the linear fitting line for extracting the sensitivity in Figure 4(d) showed more than 99.8% linearity in RCG:RSG = 1:1, 2:1, and 3:1. Figure 4(d) also shows the deviation for 20 measurements at each pH value, indicating reliable sensing performance. This means that there is no Debye screening length limit for the various pH buffer solutions used in this study.

[Comment 6]
Is there sensor to sensor variation?

[Answer 6]
Thank you for your important advice on our manuscript. We modified Figure 4(d) in the revised manuscript to demonstrate the sensor-to-sensor variation. Figure 4(d) shows the deviation for 20 measurements at each pH value, indicating reliable detection performance. 

In response to the reviewer’s comments, we have also revised the manuscript in the following section for the reader's understanding:

(Line 163 – 167, Page 5 in revised manuscript)
In addition, the linear fitting line for extracting the sensitivity in Figure 4(d) showed more than 99.8% linearity in RCG:RSG = 1:1, 2:1, and 3:1. Figure 4(d) also shows the deviation for 20 measurements at each pH value, indicating reliable sensing performance. This means that there is no Debye screening length limit for the various pH buffer solutions used in this study.

[Comment 7]
The author mentioned that "the sensor is suitable for integration with wide transmission bandwidth, high temperature, speed, and frequency power electronics.", but no direct evidence or explanation was shown in the article.

[Answer 7]
Thanks for your valuable comments. As mentioned in the introduction of the revised manuscript, HEMT devices are suitable for integration with wider transmission bandwidth, high temperature, speed and frequency power electronics compared to silicon-based transistors. Therefore, we have described the expectations of this study in the conclusion based on the characteristics of the HEMT device, but the expression has been modified as follows to avoid the reader's misunderstanding. 

In response to the reviewer’s comments, we have also revised the manuscript in the following section for the reader's understanding:

(Line 197 – 200, Page 6 in revised manuscript)
Thus, since the sensor proposed in this study is based on a HEMT device, it is expected to be suitable for integration with wide transmission bandwidth, high temperature, speed and frequency power electronics.

Again, thank you for your kind consideration and significant advices of our manuscript.
Sincerely yours,
Won¬-Ju Cho
Department of Electronic Materials Engineering, Kwangwoon University,
20, Gwangun-ro, Nowon-gu, Seoul, 01897, Republic of Korea
E-mail: chowj@kw.ac.kr

[Reference]
[R1]    Majumder, M.; Keis, K.; Zhan, X.; Meadows, C.; Cole, J.; Hinds, B. J. Enhanced electrostatic modulation of ionic diffusion through carbon nanotube membranes by diazonium grafting chemistry. J. Membr. Sci.. 2008, 316, 89-96.

Reviewer 2 Report

Please see the uploaded document.

Author Response

Response Letter to Reviewer #2 
We sincerely thank you for giving us such valuable suggestions for revision. The reviewers′ comments are indeed constructive and helpful for us to improve our manuscript. Thus, we revise our manuscript based on the reviewers′ comments exactly. The followings are our point-to-point response to the reviewers′ concerns and our descriptions on the revision, which are indicated by yellow highlight in the revised manuscript: 

[Comment 1]
Please check the simple mistakes through the manuscript, for example, “ developments in big data, artificial intelligence, deep learning, and internet of things in the fourth industrial revolution have led to increased human–machine interactions and, consequently, the requirement of the miniaturization and functionalization of electronic devices.”(Line 26), “The results of this study indicate that the pH sensor developed has high performance, stability, and reliability and is also disposable” (Line 189)…

[Answer 1]
Thank you for your helpful comment. We revised a mistake in the manuscript.

In response to the reviewer’s comments, we have also revised the manuscript in the following section for the reader's understanding:

(Line 26 – 29, Page 1 in revised manuscript)
Developments in big data, artificial intelligence, deep learning, and internet of things in the fourth industrial revolution have led to increased human–machine interactions and, consequently, the requirement of the miniaturization and functionalization of electronic devices.

(Line 196 – 197, Page 6 in revised manuscript)
The results of this study indicate that the pH sensor developed has high performance, stability, and reliability and is also disposable.

[Comment 2]
Figure legend (e.g., a, b and c) should be laid in the top left of pictures.

[Answer 2]
Chemosensors' "Microsoft Word template" states that the figure legend should be centered below the picture. I specified it according to the journal format.

[Comment 3]
The structure of the section of results needs to be reconsidered. For example, Figure 2c and Figure 4d are repeated; Figure 4 a, b, and c should be added into supporting information; authors have obtained the optimum condition (RCG:RSG=3:1), Figure 5 a and b should detect the other data.

[Answer 3]
Figure 2 is the data extracted from Silvaco TCAD simulation, and Figure 4 is the data measured in the real fabricated device. So, as the reviewer mentioned, Figure 2c (ΔVCG vs ΔVSG) and Figure 4d (ΔVCG vs pH value) are clearly different data, not repeated. Also, since RCG:RSG = 3:1 amplifies the potential to the maximum, it can be said to be an optimal condition, but in this paper, the subject is that sensitivity can be amplified in various ratios according to the mask pattern design. Therefore, in order to prevent reader misunderstanding, it is considered more appropriate to show the transfer characteristic curve of each ratio, and it would be appreciated if the reviewer understood this.

[Comment 4]
Please consider the application in different fields, determination scope, and the influence factor such as ions and temperature.

[Answer 4]
  Thank you for your kind consideration. The pH sensor proposed in this study receives the potential by using an extended gate and amplifies it by using a coplanar gate AlGaN/GaN MOS HEMT. Therefore, it is expected that when subsequent studies on the extended gate are performed, various biological events other than pH can be detected with high sensitivity. We specified this in the conclusion of the manuscript.

In response to the reviewer’s comments, we have also revised the manuscript in the following section for the reader's understanding:

(Line 200 – 204, Page 6 in revised manuscript)
In addition, the pH sensor proposed in this study detects the potential change of the sensing membrane using EG. Therefore, when the coplanar gate AlGaN/GaN MOS HEMT proposed in this study is introduced into the subsequent study on EG, various biological events such as enzyme-substrate reactions, antigen-antibody binding, and nucleic acid hybridization can be detected with high sensitivity.

Again, thank you for your kind consideration and significant advices of our manuscript.
Sincerely yours,
Won-Ju Cho
Department of Electronic Materials Engineering, Kwangwoon University,
20, Gwangun-ro, Nowon-gu, Seoul, 01897, Republic of Korea
E-mail: chowj@kw.ac.kr

Reviewer 3 Report

The paper does not demonstrated novelty enough compared with the author’s previous work (ACS Omega. 2020 Jun 9; 5(22): 12809–12815). Thus, I do not recommend it to be published in Chemosensors, at least in the present form. Nevertheless, I left a few comments.

  • The authors state that “A low gate leakage current was observed because the SiO2/Ta2O5 stacked gate insulator effectively blocked the gate leakage and suppressed the surface leakage.” But some leakage can be seen in Fig. 3a. Regardless, and considering that leakage mechanisms are not being taken into account in the TCAD simulation, there is no apparent advantage in using this type of simulation for predicting deltaVcg, as it is simply predictable from equations (1-3) as the authors note. Also, there is no information in the TCAD simulation, nor direct comparison with the experimental transfer curves.
  • It would be interesting to show the sensibility of the sensor layer individually (i.e. deltaVsg VS delta_ph). This would allow to compare to the sensibility obtained with the explored architecture.

Author Response

Response Letter to Reviewer #3 
We sincerely thank you for giving us such valuable suggestions for revision. The reviewers′ comments are indeed constructive and helpful for us to improve our manuscript. Thus, we revise our manuscript based on the reviewers′ comments exactly. The followings are our point-to-point response to the reviewers′ concerns and our descriptions on the revision, which are indicated by yellow highlight in the revised manuscript: 

[Comment 1]
The paper does not demonstrated novelty enough compared with the author’s previous work (ACS Omega. 2020 Jun 9; 5(22): 12809–12815). Thus, I do not recommend it to be published in Chemosensors, at least in the present form. Nevertheless, I left a few comments. The authors state that “A low gate leakage current was observed because the SiO2/Ta2O5 stacked gate insulator effectively blocked the gate leakage and suppressed the surface leakage.” But some leakage can be seen in Fig. 3a. Regardless, and considering that leakage mechanisms are not being taken into account in the TCAD simulation, there is no apparent advantage in using this type of simulation for predicting deltaVcg, as it is simply predictable from equations (1-3) as the authors note. Also, there is no information in the TCAD simulation, nor direct comparison with the experimental transfer curves. It would be interesting to show the sensibility of the sensor layer individually (i.e. deltaVsg VS deltaph). This would allow to compare to the sensibility obtained with the explored architecture.

[Answer 1]
Thank you for your valuable comment. However, there seems to be a significant misunderstanding of the reviewer about this study, so we will explain it in detail. First of all, there is a clear difference from the work of our previous work you mentioned (ACS Omega. 2020 Jun 9; 5(22): 12809–12815) and this manuscript submitted to Chemosensors. Our previous work you mentioned is a silicon-based device that amplifies the sensitivity through capacitive coupling between the coplanar gate and the floating gate. Here, the capacitive coupling effect and sensitivity amplification according to the coplanar gate area were investigated. On the other hand, this manuscript submitted to Chemosensors is a HEMT-based device that amplifies the sensitivity through resistive coupling between the control gate and the sensing gate. In particular, the resistive coupling effect and the sensitivity amplification according to the ratio of the resistive layer connected to the coplanar gate and the sensing gate were investigated. 
The transfer characteristic curve and the gate current at this time of the coplanar gate AlGaN/GaN MOS HEMT proposed in this study are shown in Figure 3(a) of the revised manuscript. Compared with the references, the device presented in this study is considered to have a sufficiently low gate leakage current due to the MOS structure [R1-R3]. 

Figure R1. Comparison of Conventional HEMT and MOS-HEMT [R2]

The Silvaco TCAD simulation proposed in this study was not performed for specific comparisons with the electrical properties of the fabricated devices, but was to evaluate the signal amplification capability by resistive coupling. In other words, what we were looking for in this simulation was the threshold voltage shift, not the leakage current. Therefore, we conducted the TCAD simulation as follows.
1) Atals simulation for MOS HEMT device with conmob (specify concentration dependent mobility model), fldmob (specify lateral electric field-dependent model), SRH (specifies Shockley-Read-Hall recombination). 
2) Mixedmode module in Atlas for circuit simulation for resistive coupling between CG and SG. 
3) Tunneling conduction such as band-to-band tunneling, Fowler-Nordheim tunneling, direct quantum tunneling, trap assist tunneling, phonon assist electron tunneling, Schottky tunneling, etc. were ignored. 
We would appreciate it if the reviewer noticed that the goal of Silvaco TCAD simulation was not to directly compare electrical characteristics with the manufactured device or to identify the leakage mechanism. Rather, we aimed to predict sensor sensitivity amplification.

Again, thank you for your kind consideration and significant advices of our manuscript.

Sincerely yours,
Won-Ju Cho
Department of Electronic Materials Engineering, Kwangwoon University,
20, Gwangun-ro, Nowon-gu, Seoul, 01897, Republic of Korea
E-mail: chowj@kw.ac.kr[Reference]
[R1]    Freedsman, J. J.; Kubo, T.; Egawa, T. High Drain Current Density E-Mode Al2O3/AlGaN/GaN MOS-HEMT on Si With Enhanced Power Device Figure-of-Merit (4×108 V2Ω-1cm-2). IEEE Trans. Electron Devices, 2013, 60, 3079-3083.
[R2]    Seok, O.; Ahn, W.; Han, M. K.; Ha, M. W. High on/off current ratio AlGaN/GaN MOS-HEMTs employing RF-sputtered HfO2 gate insulators. Semicond. Sci. Technol. 2012, 28, 025001.
[R3]    Freedsman, J. J.; Egawa, T.; Yamaoka, Y.; Yano, Y.; Ubukata, A.; Tabuchi, T.; Matsumoto, K. Normally-off Al2O3/AlGaN/GaN MOS-HEMT on 8 in. Si with low leakage current and high breakdown voltage (825 V). Appl. Phys. Express. 2014, 7, 041003.

Round 2

Reviewer 1 Report

Accept.

Author Response

Thank you for your kind consideration.

Sincerely yours,
Won-Ju Cho
Department of Electronic Materials Engineering, Kwangwoon University,
20, Gwangun-ro, Nowon-gu, Seoul, 01897, Republic of Korea
E-mail: chowj@kw.ac.kr

Reviewer 2 Report

Accept in present form

Author Response

(The authors gave the same response as above.)

Reviewer 3 Report

The authors clarified a few points, but I still have some questions/comments.

  • The authors explained in the response letter how were conducted the TCAD simulations, however I think this should be better explained also in the manuscript.
  • Since the authors have two similar methods, even using different materials, I think this should be added to the introduction as literature review.

Author Response

Response Letter to Reviewer #3

We sincerely thank you for giving us such valuable suggestions for revision. The reviewers′ comments are indeed constructive and helpful for us to improve our manuscript. Thus, we revise our manuscript based on the reviewers′ comments exactly. The followings are our point-to-point response to the reviewers′ concerns and our descriptions on the revision, which are indicated by yellow highlight in the revised manuscript: 

[Comment 1]

The authors clarified a few points, but I still have some questions/comments. The authors explained in the response letter how were conducted the TCAD simulations, however I think this should be better explained also in the manuscript. Since the authors have two similar methods, even using different materials, I think this should be added to the introduction as literature review.

[Answer 1]

Thank you for your important advice on our manuscript. As a reference to your advice, we have described the conditions of Silvaco TCAD simulation in the revised manuscript. In addition, we have provided information about our previous work in the Introduction of the revised manuscript so that readers do not misunderstand our two similar methods.

In response to the reviewer’s comments, we have also revised the manuscript in the following section for the reader's understanding:

(Line 36 – Line 40, Page 1 in revised manuscript)

In addition, in our previous work, we studied a silicon-based device that amplifies the sensitivity through capacitive coupling between the coplanar gate and the floating gate, and here we investigated the capacitive coupling effect according to the coplanar gate area and the corresponding sensitivity amplification [9].

(Line 122, Page 3 – Line 130, Page 4 in revised manuscript)

The Silvaco TCAD simulation proposed in this study is to evaluate the signal amplifica-tion capability by resistive coupling, and was performed as follows: 1) Atals simulation for MOS HEMT device with conmob (specify concentration dependent mobility model), fldmob (specify lateral electric field-dependent model), SRH (specifies Shockley-Read-Hall recombination). 2) Mixedmode module in Atlas for circuit simulation for resistive cou-pling between CG and SG. 3) Tunneling conduction such as band-to-band tunneling, Fowler-Nordheim tunneling, direct quantum tunneling, trap assist tunneling, phonon as-sist electron tunneling, Schottky tunneling, etc. were ignored.

Again, thank you for your kind consideration and significant advices of our manuscript.

Sincerely yours,

Won­-Ju Cho

Department of Electronic Materials Engineering, Kwangwoon University,

20, Gwangun-ro, Nowon-gu, Seoul, 01897, Republic of Korea

E-mail: chowj@kw.ac.kr
